# Differentially Private Learning with Adaptive Clipping

**Galen Andrew**
galenandrew@google.com

**Om Thakkar**
omthkkr@google.com

**H. Brendan McMahan**
mcmahan@google.com

**Swaroop Ramaswamy**
swaroopram@google.com

## Abstract

Existing approaches for training neural networks with user-level differential privacy (e.g., DP Federated Averaging) in federated learning (FL) settings involve bounding the contribution of each user's model update by *clipping* it to some constant value. However there is no good *a priori* setting of the clipping norm across tasks and learning settings: the update norm distribution depends on the model architecture and loss, the amount of data on each device, the client learning rate, and possibly various other parameters. We propose a method wherein instead of a fixed clipping norm, one clips to a value at a specified quantile of the update norm distribution, where the value at the quantile is itself estimated online, with differential privacy. The method tracks the quantile closely, uses a negligible amount of privacy budget, is compatible with other federated learning technologies such as compression and secure aggregation, and has a straightforward joint DP analysis with DP-FedAvg. Experiments demonstrate that adaptive clipping to the median update norm works well across a range of realistic federated learning tasks, sometimes outperforming even the best fixed clip chosen in hindsight, and without the need to tune any clipping hyperparameter.

## 1   Introduction

Deep learning has become ubiquitous, with applications as diverse as image processing, natural language translation, and music generation [27, 13, 28, 7]. Deep models are able to perform well in part due to their ability to utilize vast amounts of data for training. However, recent work has shown that it is possible to extract information about individual training examples using only the parameters of a trained model [12, 30, 26, 8, 19]. When the training data potentially contains privacy-sensitive user information, it becomes imperative to use learning techniques that limit such memorization.

Differential privacy (DP) [10, 11] is widely considered a gold standard for bounding and quantifying the privacy leakage of sensitive data when performing learning tasks. Intuitively, DP prevents an adversary from confidently making any conclusions about whether some user's data was used in training a model, even given access to arbitrary side information. The formal definition of DP depends on the notion of neighboring datasets: we will refer to a pair of datasets $D, D' \in \mathcal{D}$ as neighbors if $D'$ can be obtained from $D$ by adding or removing one element.

**Definition 1.1** (Differential Privacy). *A (randomized) algorithm $M : \mathcal{D} \to \mathcal{R}$ with input domain $\mathcal{D}$ and output range $\mathcal{R}$ is $(\varepsilon, \delta)$-differentially private if for all pairs of neighboring datasets $D, D' \in \mathcal{D}$, and every measurable $S \subseteq \mathcal{R}$, we have $\Pr(M(D) \in S) \leq e^{\varepsilon} \cdot \Pr(M(D') \in S) + \delta$, where probabilities are with respect to the coin flips of $M$.*

35th Conference on Neural Information Processing Systems (NeurIPS 2021).

Following McMahan et al. [18], we define two common settings of privacy corresponding to two different definitions of neighboring datasets. In *example-level* DP, datasets are considered neighbors when they differ by the addition or removal of a single example [9, 4, 2, 22, 31, 23, 15]. In *user-level* DP, neighboring datasets differ by the addition or removal of *all of the data of one user* [18]. User-level DP is the stronger form, and is preferred when one user may contribute many training examples to the learning task, as privacy is protected even if the same privacy-sensitive information occurs in all the examples from one user. In this paper, we will describe the technique and perform experiments in terms of the stronger user-level form, but we note that example-level DP can be achieved by simply giving each user a single example.

To achieve user-level DP, we employ the Federated Averaging algorithm [17], introduced as a decentralized approach to model training in which the training data is left distributed on user devices, and each training round aggregates updates that are computed locally. On each round, a sample of devices are selected for training, and then each selected device performs potentially many steps of local SGD over minibatches of its own data, sending back the model delta as its update.

Bounding the influence of any user in Federated Averaging is both necessary for privacy and often desirable for stability. One popular way to do this to cap the $L_2$ norm of its model update by projecting larger updates back to the ball of norm $C$. Since such clipping also effectively bounds the $L_2$ sensitivity of the aggregate with respect to the addition or removal of any user's data, adding Gaussian noise to the aggregate is sufficient to obtain a central differential privacy guarantee for the update [9, 4, 2]. Standard composition techniques can then be used to extend the per-update guarantee to the final model [20].

Setting an appropriate value for the clipping threshold $C$ is crucial for the utility of the private training mechanism. Setting it too low can result in high bias since we discard information contained in the magnitude of the gradient. However setting it too high entails the addition of more noise, because the amount of Gaussian noise necessary for a given level of privacy must be proportional to the norm bound (the $L_2$ sensitivity), and this will eventually destroy model utility. The clipping bias-variance trade-off was observed empirically by McMahan et al. [18], and was theoretically analyzed and shown to be an inherent property of differentially private learning by Amin et al. [3].

Learning large models using the Federated Averaging/SGD algorithm [17, 18] can take thousands of rounds of interaction between the central server and the clients. The norms of the updates can vary as the rounds progress. Prior work [18] has shown that decreasing the value of the clipping threshold after training a language model for some initial number of rounds can improve model accuracy. However, the behavior of the norms can be difficult to predict without prior knowledge about the system, and if it is difficult to choose a fixed clipping norm for a given learning task, it is even more difficult to choose a parameterized clipping norm schedule.

While there has been substantial work on DP techniques for learning, almost every technique has hyperparameters which need to be set appropriately for obtaining good utility. Besides the clipping norm, learning techniques have other hyperparameters which might interact with privacy hyperparameters. For example, the server learning rate in DP SGD might need to be set to a high value if the clipping threshold is very low, and vice-versa. Such tuning for large networks can have an exorbitant cost in computation and efficiency, which can be a bottleneck for real-world systems that involve communicating with millions of samples for training a single network. Tuning may also incur an additional cost for privacy, which needs to be accounted for when providing a privacy guarantee for the released model with tuned hyperparameters (though in some cases, hyperparameters can be tuned using the same model and algorithm on a sufficiently-similar public proxy dataset).

**Related Work** The heuristic of clipping to the median update norm was suggested by Abadi et al. [1], but no method for doing so privately was proposed, nor was the heuristic empirically tested. One possibility would be to estimate the median unclipped update norm at each round privately by adding noise calibrated to the smooth sensitivity as defined by Nissim et al. [21]. However this approach has several drawbacks compared to the algorithm we will present. It would require the clients to send their unclipped updates to the server at each round,[1] which is incompatible with the foundational

---

[1]Alternatively, clients could first report their update norms, after which the server could reply with the clipping norm for the round, and finally the clients would send the clipped updates. However, this would still give the server strictly more information than our approach, and would require an extra round of communication, both of which are undesirable.

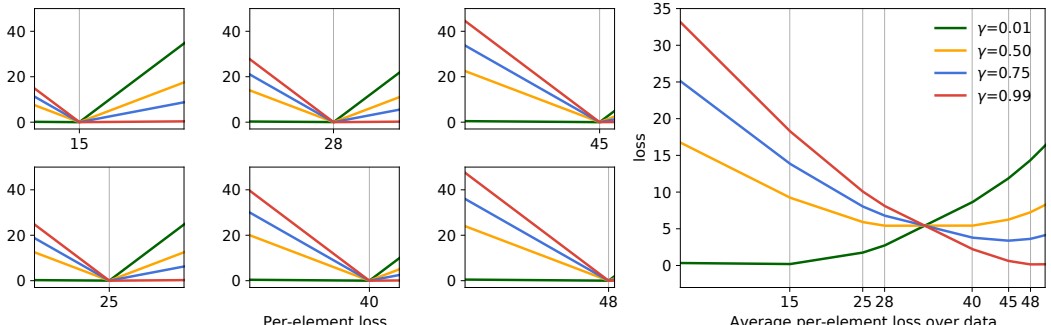

Figure 1: Loss functions to estimate the 0.01-, 0.5-, 0.75-, and 0.99-quantiles for a random variable $X$ that uniformly takes values in $\{15, 25, 28, 40, 45, 48\}$. The loss function is the average of convex piecewise-linear functions, one for each value. For instance, for the median ($\gamma = 0.5$), this is just $\ell_\gamma(C; X) = \frac{1}{2}|X - C|$, where $X$ is the random value, and $C$ is the estimate. When we average these functions, we arrive at the yellow function in the plot showing the average loss, which indeed is minimized by any value between the central two elements, i.e., in the interval $[28, 40]$. The function for $\gamma = 0.75$ is minimized at $C = 45$ because $\Pr(X \leq C) < 0.75$ for values $C$ in $[40, 45)$, while $\Pr(X \leq C) > 0.75$ for values $C$ in $(45, 48]$.

principle of federated learning to transmit only focused, minimal updates, and precluding the use of secure aggregation [5] and certain forms of compression [16, 6]. Also, by incorporating information across multiple rounds, our method is able to track the underlying quantile closely, with less jitter associated with sampling at each round, and using only a negligible fraction of the privacy budget.

**Contributions** In this paper, we describe a method for adaptively and privately tuning the clipping threshold to track a given quantile of the update norm distribution during training. The method uses a negligible amount of privacy budget, and is compatible with other FL technologies such as compression and secure aggregation [5, 6]. We perform a careful empirical comparison of our adaptive clipping method to a highly optimized fixed-clip baseline on a suite of realistic and publicly available FL tasks to demonstrate that high-utility and high-privacy models—sometimes exceeding any fixed clipping norm in utility—can be trained using our method without the need to tune any clipping hyperparameter.

## 2 Private adaptive quantile clipping

In this section, we will describe the adaptive strategy that can be used for adjusting the clipping threshold so that it comes to approximate the value at a specified quantile.

Let $X \in \mathbb{R}$ be a random variable, let $\gamma \in [0, 1]$ be a quantile to be matched. For any $C$, define

$$\ell_\gamma(C; X) = \begin{cases} (1 - \gamma)(C - X) & \text{if } X \leq C, \\ \gamma(X - C) & \text{otherwise,} \end{cases}$$

so

$$\ell'_\gamma(C; X) = \begin{cases} (1 - \gamma) & \text{if } X \leq C, \\ -\gamma & \text{otherwise.} \end{cases}$$

Hence, $\mathbb{E}[\ell'_\gamma(C; X)] = (1 - \gamma)\Pr[X \leq C] - \gamma\Pr[X > C] = \Pr[X \leq C] - \gamma$. For $C^*$ such that $\mathbb{E}[\ell'_\gamma(C^*; X)] = 0$, we have $\Pr[X \leq C^*] = \gamma$. Thus, $C^*$ is the $\gamma^{\text{th}}$ quantile of $X$. Because the loss is convex and has gradients bounded by 1, we can get an online estimate of $C$ that converges to the $\gamma^{\text{th}}$ quantile of $X$ using online gradient descent (see, e.g., Shalev-Shwartz [25]). See Figure 1 for a plot of the loss for a random variable that takes six values with equal probability.

Suppose at some round we have $m$ samples of $X$, with values $(x_1, \ldots, x_m)$. The average derivative of the loss for that round is

$$\bar{\ell}'_\gamma(C; X) = \frac{1}{m} \sum_{i=1}^{m} \begin{cases} (1-\gamma) & \text{if } x_i \leq C, \\ -\gamma & \text{otherwise} \end{cases}$$

$$= \frac{1}{m} \left( (1-\gamma) \sum_{i \in [m]} \mathbb{I}_{x_i \leq C} - \gamma \sum_{i \in [m]} \mathbb{I}_{x_i > C} \right) = \bar{b} - \gamma,$$

where $\bar{b} \triangleq \frac{1}{m} \sum_{i \in [m]} \mathbb{I}_{x_i \leq C}$ is the empirical fraction of samples with value at most $C$. For a given learning rate $\eta_C$, we can perform the update: $C \leftarrow C - \eta_C(\bar{b} - \gamma)$.

**Geometric updates.** Since $\bar{b}$ and $\gamma$ take values in $[0, 1]$, the linear update rule above changes $C$ by a maximum of $\eta_C$ at each step. This can be slow if $C$ is on the wrong order of magnitude. At the other extreme, if the optimal value of $C$ is orders of magnitude smaller than $\eta_C$, the update can be very coarse, and may overshoot to become negative. To remedy such issues, we propose the following geometric update rule: $C \leftarrow C \cdot \exp(-\eta_C(\bar{b} - \gamma))$. This update rule converges quickly to the true quantile even if the initial estimate is off by orders of magnitude. It also has the attractive property that the variance of the estimate around the true quantile at convergence is proportional to the value at that quantile. In our experiments, we use the geometric update rule with $\eta_C = 0.2$.

## 2.1 DP-FedAvg with adaptive quantile clipping

Let $m$ be the number of users in a round and let $\gamma \in [0, 1]$ denote the target quantile of the norm distribution at which we want to clip. For iteration $t \in [T]$, let $C^t$ be the clipping threshold, and $\eta_C$ be the learning rate. Let $\mathcal{Q}^t$ be set of users sampled in round $t$. Each user $i \in \mathcal{Q}^t$ will send the bit $b_i^t$ along with the usual model delta update $\Delta_i^t$, where $b_i^t = \mathbb{I}_{||\Delta_i^t||_2 \leq C^t}$. Defining $\bar{b}^t = \frac{1}{m} \sum_{i \in \mathcal{Q}^t} b_i^t$, we would like to apply the update $C \leftarrow C \cdot \exp(-\eta_C(\bar{b} - \gamma))$. However, we can't use $\bar{b}^t$ directly, since it may reveal private information about the magnitude of users' updates. To remedy this, we add Gaussian noise to the sum: $\tilde{b}^t = \frac{1}{m} \left( \sum_{i \in \mathcal{Q}^t} b_i^t + \mathcal{N}(O, \sigma_b^2) \right)$. The DPFedAvg algorithm with adaptive clipping is shown in Algorithm 1. We augment basic federated averaging with server momentum, which improves convergence [14, 24].[2]

---

**Algorithm 1** DPFedAvg-M with adaptive clipping

---

**function** Train($m, \gamma, \eta_c, \eta_s, \eta_C, z, \sigma_b, \beta$)
  Initialize model $\theta^0$, clipping bound $C^0$
  $z_\Delta \leftarrow \left( z^{-2} - (2\sigma_b)^{-2} \right)^{-1/2}$
  **for** each round $t = 0, 1, 2, \ldots$ **do**
    $\mathcal{Q}^t \leftarrow$ (sample $m$ users uniformly)
    **for** each user $i \in \mathcal{Q}^t$ **in parallel do**
      $(\Delta_i^t, b_i^t) \leftarrow$ FedAvg($i, \theta^t, \eta_c, C^t$)
    $\sigma_\Delta \leftarrow z_\Delta C^t$
    $\tilde{\Delta}^t = \frac{1}{m} \left( \sum_{i \in \mathcal{Q}^t} \Delta_i^t + \mathcal{N}(0, I\sigma_\Delta^2) \right)$
    $\bar{\Delta}^t = \beta \bar{\Delta}^{t-1} + (1-\beta)\tilde{\Delta}^t$
    $\theta^{t+1} \leftarrow \theta^t + \eta_s \bar{\Delta}^t$
    $\tilde{b}^t = \frac{1}{m} \left( \sum_{i \in \mathcal{Q}^t} b_i^t + \mathcal{N}(O, \sigma_b^2) \right)$

$C^{t+1} \leftarrow C^t \cdot \exp\left( -\eta_C(\tilde{b}^t - \gamma) \right)$

**function** FedAvg($i, \theta^0, \eta, C$)
  $\theta \leftarrow \theta^0$
  $\mathcal{G} \leftarrow$ (user $i$'s local data split into batches)
  **for** batch $g \in \mathcal{G}$ **do**
    $\theta \leftarrow \theta - \eta \nabla \ell(\theta; g)$
  $\Delta \leftarrow \theta - \theta^0$
  $b \leftarrow \mathbb{I}_{||\Delta|| \leq C}$
  $\Delta' \leftarrow \Delta \cdot \min\left( 1, \frac{C}{||\Delta||} \right)$
  **return** $(\Delta', b)$

---

[2]Note that since the momentum update is computed using privatized estimates of the average client delta, privacy properties are unchanged when momentum is added. We also experimented with adaptive learning rates, but found that they were less effective when noise is added for DP, perhaps because the noise causes the preconditioner $v_t$ to become large prematurely.

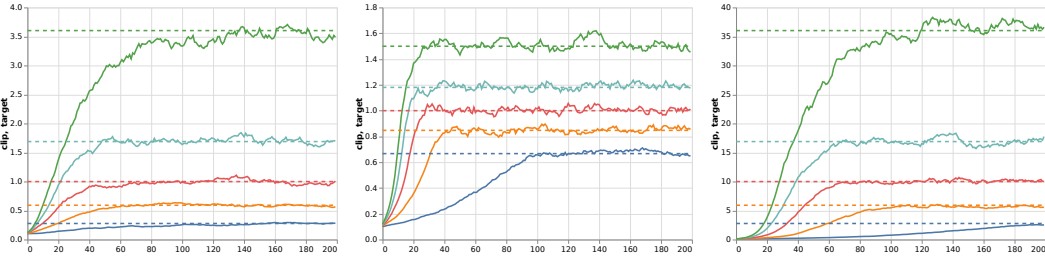

Figure 2: Evolution of the quantile estimate on data drawn from log-normal distributions. The three plots use data drawn from the exponential of $\mathcal{N}(0.0, 1.0)$, $\mathcal{N}(0.0, 0.1)$, and $\mathcal{N}(\log 10, 1.0)$, respectively. Curves are shown for each of five quantiles: $(0.1, 0.3, 0.5, 0.7, 0.9)$, and the dashed lines show the true value at each quantile. Hyperparameters are as discussed in the text and used in the experiments of Section 3: $\eta_C = 0.2, C^0 = 0.1, m = 100, \sigma_b = m/20$. After an initial phase of exponential growth, the true quantile is fairly closely tracked. A smaller value of $\eta_C$ would allow more accurate tracking at the cost of slower convergence, but since the quantile value is only used as a heuristic for clipping, a small amount of noise is tolerable. The entire sequence of values estimated for each target quantile satisfy $(0.034, n^{-1.1})$-differential privacy using RDP composition across the 200 rounds assuming fixed-size samples of $m = 100$ out of a total population of $n = 10^6$ [29].

**Theorem 1.** *One step of DP-FedAvg with adaptive clipping using $\sigma_b$ noise standard deviation on the clipped counts $\sum b_i^t$ and $z_\Delta$ noise multiplier on the vector sums $\sum \Delta_i^t$ is equivalent (so far as privacy accounting is concerned) to one step of non-adaptive DP-FedAvg with noise multiplier $z$ if we set $z_\Delta = \left(z^{-2} - (2\sigma_b)^{-2}\right)^{-1/2}$.*

*Proof.* We make a conceptual change to the algorithm that does not change the behavior or privacy properties but allows us to analyze each step as if it were a single private Gaussian sum. Instead of sending $(\Delta_i^t, b_i^t)$, each user sends $(\hat{\Delta}_i^t, \hat{b}_i^t) \triangleq \left(\Delta_i^t/\sigma_\Delta, (b_i^t - 1/2)/\sigma_b\right)$. The server adds noise with covariance $I$ and averages, then reverses the transformation so $\tilde{\Delta}^t = \frac{\sigma_\Delta}{m}\left(\sum_{i \in \mathcal{Q}^t} \hat{\Delta}_i^t + \mathcal{N}(0, I)\right)$ and $\tilde{b}^t = \frac{\sigma_b}{m}\left(\sum_{i \in \mathcal{Q}^t} \hat{b}_i^t + \mathcal{N}(0, 1)\right) + 1/2$. Noting that $||(\hat{\Delta}_i^t, \hat{b}_i^t)|| \leq S \triangleq \left((C^t/\sigma_\Delta)^2 + (1/2\sigma_b)^2\right)^{1/2}$, it is clear that the two Gaussian sum queries of Algorithm 1 are equivalent to pre- and post-processing of a single query with sensitivity $S$ and covariance $I$, or noise multiplier $z = 1/S = \left(z_\Delta^{-2} + (2\sigma_b)^{-2}\right)^{-1/2}$. Rearranging yields the result. $\square$

In practice, we recommend using a value of $\sigma_b = m/20$. Since the noise is Gaussian, this implies that the error $|\tilde{b}^t - \bar{b}^t|$ will be less than 0.1 with 95.4% probability, and will be no more than 0.15 with 99.7% probability. Even in this unlikely case, assuming a geometric update and a learning rate of $\eta_C = 0.2$, the error on the update would be a factor of $\exp(0.2 \times 0.15) = 1.03$, a small deviation. So this default gives high privacy for an acceptable amount of noise in the quantile estimation process. Using Thm. 1, we can compute that to achieve an effective combined noise multiplier of $z = 1$, with $m = 100$ clients per round, the noise multiplier $z_\Delta$ is approximately 1.005. So we are paying only a factor of 0.5% more noise on the updates for adaptive clipping with the same privacy guarantee (a quantity which only gets smaller with increasing $m$). These constants ($\sigma_b = m/20$ and $\eta_C = 0.2$) are what we use in the experiments of Section 3.

The clipping norm can be initialized to any value $C^0$ that is safely on the low end of the expected norm distribution. If it is too high and needs to adapt downward, a lot of noise may be added at the beginning of model training, which may swamp the model. However there is little danger in setting it quite low, since the geometric update will make it grow exponentially until it matches the true quantile. In our experiments we use an initial clip of 0.1 for all tasks. It is easy to compute that with a learning rate of $\eta_C = 0.2$ and a target quantile of $\gamma = 0.5$, if every update is clipped, the quantile estimate will increase by a factor of ten every 23 iterations. In order to show the effectiveness of the algorithm at tracking a known quantile, we ran it on simulated data for which we can compute the true quantile exactly. Figure 2 shows the result of this experiment.

| Task | model | $N$ | $n$ | $T$ | $z$ | $m$ | $\eta_c$ | $\eta_s$ | $C_{\min}$ | $C_{\max}$ |
|------|-------|-----|-----|-----|-----|-----|----------|----------|------------|------------|
| CIFAR-100 | ResNet | 11M | 500 | 4000 | 0.669 | 2231 | 0.1 | 0.32 | 0.75 | 2.2 |
| EMNIST-CR | CNN | 1.2M | 3400 | 1500 | 0.513 | 513 | 0.032 | 1.0 | 0.28 | 0.85 |
| EMNIST-AE | Deep AE | 2.8M | 3400 | 3000 | 0.659 | 2197 | 3.2 | 1.78 | 0.22 | 0.95 |
| SHAKESPEARE | C-LSTM | 820k | 715 | 1200 | 0.510 | 510 | 1.0 | 0.32 | 0.25 | 3.6 |
| SO-NWP | W-LSTM | 4.1M | 342k | 1500 | 1.396 | 13958 | 0.18 | 1.78 | 0.30 | 1.6 |
| SO-LR | Multi-LR | 5M | 342k | 1500 | 1.396 | 13958 | 320.0 | 1.78 | 16.0 | 135.0 |

Table 1: Dataset statistics and chosen hyperparameters. Left: model type, number of trainable parameters $N$, number of training clients $n$, and the number of training rounds $T$ used (following Reddi et al. [24]). Middle: the noise multiplier $z$ and number of clients per round $m$ necessary to achieve $(5, n^{-1.1})$-DP with less than 5% model performance loss if each task had a population of $n = 10^6$ [29]. Right: the optimal unclipped baseline client and server learning rates (Sec. 3.1) for each task and chosen values of minimum and maximum fixed clips (Sec. 3.2).

## 3 Experiments

To empirically validate the approach, we examine the behavior of our algorithm on six of the public benchmark federated learning tasks defined by Reddi et al. [24], which are to our knowledge the most realistic and representative publicly available federated learning tasks that exist to date. All six tasks are non-i.i.d. with respect to user partitioning: indeed with the exception of CIFAR-100, the data is partitioned according to the actual human user who generated the data, for example the writer of the EMNIST characters or the Stack Overflow user who asked or answered a question. Table 1 (left) lists the characteristics of the datasets.

Two of the tasks derived from Stack Overflow data (SO-NWP and SO-LR) are ideal for DP research due to the very high number of users (342k) making it possible to train models with good user-level privacy without sacrificing accuracy. The other four tasks (CIFAR-100, EMNIST-AE, EMNIST-CR, SHAKESPEARE) are representative learning tasks, but not representative population sizes for real world cross-device FL applications. Therefore we focus on establishing that adaptive clipping works well with 100 clients per round on these tasks in the regime where the noise is at a level such that utility is just beginning to degrade. Under the assumption that a larger population were available, one could increase $m$, $\sigma_\Delta$, and $\sigma_b$ proportionally to achieve comparable utility with high privacy. This should not significantly affect convergence (indeed, it might be beneficial) since the only effect is to increase the number of users in the average $\tilde{\Delta}^t$, reducing the variance. Table 1 (middle) shows the number of clients per round with which our experiments indicate we could achieve $(5, n^{-1.1})$-DP for each dataset with acceptable model performance loss (less than 5% relative to non-private training, as discussed later) if each dataset had $n = 10^6$ clients, using RDP composition with fixed-size subsampling [29].[3]

### 3.1 Baseline client and server learning rates

Reddi et al. [24] provide optimized client and server learning rates for federated averaging with momentum that serve as a starting point for our experimental setup. For almost all hyperparameters (model configuration, evaluation metrics, client batch size, total rounds, etc.) we replicate their experiments, but with two changes. First, we increase the number of clients per round to 100 for all tasks. This reduces the variance in the updates to a level where we can reasonably assume that adding more clients is unlikely to significantly change convergence properties [18]. Second, as shown in Algorithm 1 we use *unweighted* federated averaging, thus eliminating the need to set yet another difficult-to-fit hyperparameter: the expected total weight of clients in a round. Since these changes might require different settings, we reoptimize the client and server learning rates for our baseline with no clipping or noise. We ran a small grid of 25 configurations for each task jointly exploring client and server learning rates whose logarithm (base-10) differs from the values in Table 10 of Reddi et al. [24] by { -½, -¼, 0, ¼, ½ }. The optimal baseline client and server learning rates for our experimental setup are shown in Table 1 (right).

---

[3]The code used for all of our experiments is publicly available at `https://github.com/google-research/federated/blob/master/differential_privacy/run_federated.py`.

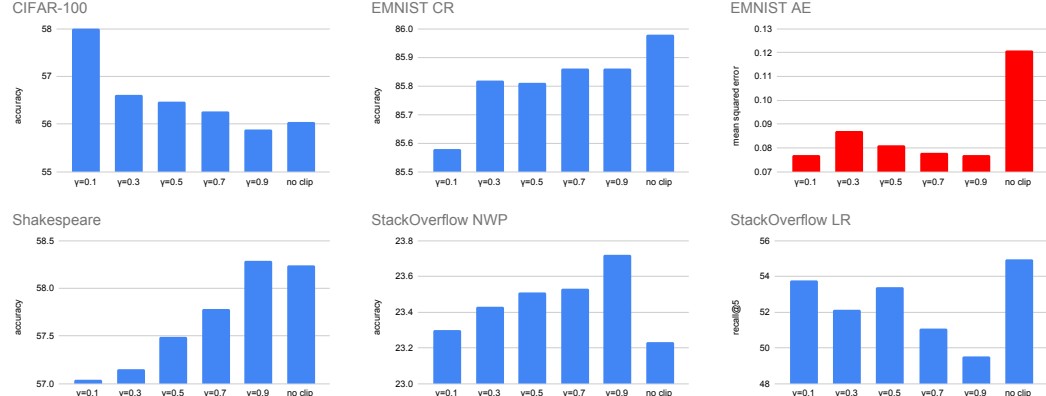

Figure 3: **Impact of clipping without noise.** Performance of the unclipped baseline compared to five settings of $\gamma$, from $\gamma = 0.1$ (aggressive clipping) to $\gamma = 0.9$ (mild clipping). The values shown are the evaluation metrics on the validation set averaged over the last 100 rounds. Note that the $y$-axes have been compressed to show small differences, and that for EMNIST-AE lower values are better.

Because clipping (whether fixed or adaptive) reduces the average norm of the client updates, it may be necessary to use a higher server learning rate to compensate. Therefore, for all approaches with clipping—fixed or adaptive—we search over a small grid of five server learning rates, scaling the values in Table 1 by $\{1, 10^{1/4}, 10^{1/2}, 10^{3/4}, 10\}$. For all configurations, we report the best performing model whose server learning rate was chosen from this small grid on the validation set.[4]

We first examine the impact of adaptive clipping without noise to see how it affects model performance. Figure 3 compares baseline performance without clipping to adaptive clipping with five different quantiles. For each quantile, we show the best model after tuning over the five server learning rates mentioned above on the validation set. On three tasks (CIFAR-100, EMNIST-AE, SO-NWP), clipping improves performance relative to the unclipped baseline. On SHAKESPEARE and SO-LR performance is slightly worse, but we can conclude that adaptive clipping to the median generally fares well compared to not using clipping across tasks. Note that for our primary goal of training with DP, it is essential to limit the sensitivity one way or another, so the modest decrease in performance observed from clipping on some tasks may be part of the inevitable tension between privacy and utility.

## 3.2 Fixed-clip baselines

We would like to compare our adaptive clipping approach to a fixed clipping baseline, but comparing to just one fixed-clip baseline may not be enough to demonstrate that adaptive clipping consistently performs well. Instead, our strategy will be to show that quantile-based adaptive clipping performs as well or nearly as well as *any* fixed clip chosen in hindsight. If we can first identify clipping norms that span the range of normal values during training on each problem/configuration, we can compare adaptive clipping to fixed clipping with those norms.

To that end, we first use adaptive clipping without noise to discover the value of the update norm distribution at the following five quantiles: $\{0.1, 0.3, 0.5, 0.7, 0.9\}$. Then we choose as the minimum of our fixed clipping range the smallest value at the 0.1 quantile over the course of training, and as the maximum the largest value at the 0.9 quantile. Plots of the update norms during training on each of the tasks are shown in Figure 4.

On each task there is a ramp up period where the clipping norm, initialized to 0.1 for all tasks, catches up to the correct norm distribution. Thus we disregard norm values collected until the actual fraction of clipped counts $\bar{b}^t$ on some round is within 0.05 of the target quantile $\gamma$. The chosen values for the

---

[4]Note that this modest retuning of the server learning rate is only necessary because we are starting from a configuration that was optimized without clipping. In practice, as we will discuss in Section 4, we recommend that all hyperparameter optimization should be done with adaptive clipping enabled from the start, eliminating the need for this extra tuning.

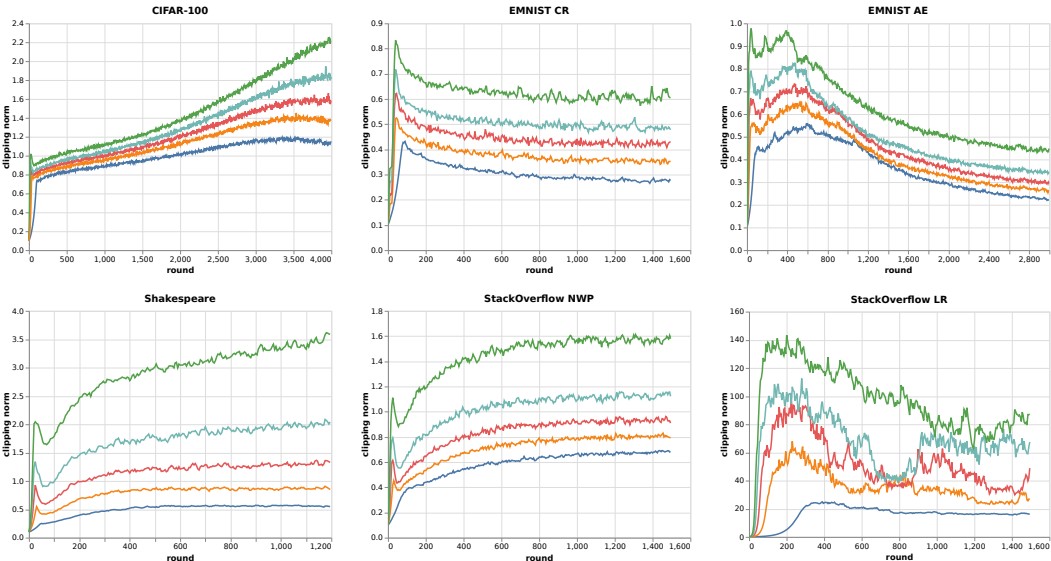

Figure 4: Evolution of the adaptive clipping norm at five different quantiles (0.1, 0.3, 0.5, 0.7, 0.9) on each task with no noise. The norms are estimated using geometric updates with $\eta_C = 0.2$ and an initial value $C^0 = 0.1$. With the possible exception of SO-LR, the estimated quantiles appear to closely track an evolving update norm distribution. Note that each task has a unique shape to its update norm evolution, which further motivates an adaptive approach.

minimum and maximum fixed clips for each task are shown in Table 1 (right). Our fixed-clipping baseline uses five fixed clipping norms logarithmically spaced in that range. Here we are taking advantage of having already run adaptive clipping to minimize the number of fixed clip settings we need to explore for each task. If we had to explore over the entire range knowing only the endpoints across all tasks (0.22, 135.0) at the same resolution, we would need nearly four times as many clip values per task.

For each value of noise multiplier $z \in \{0, 0.01, 0.03, 0.1\}$ we trained using the five fixed clipping norms and compare to adaptive clipping with the five quantiles (0.1, 0.3, 0.5, 0.7, 0.9). Note that for the fixed clipping runs $z_\Delta = z$; that is, for fixed clip $C$, the noise applied to the the updates has standard deviation $zC$. As discussed in section 2.1, on the adaptive clipping runs $z_\Delta$ is slightly higher due to the need to account for privacy when estimating the clipped counts.

### 3.3 Comparison of fixed and adaptive clipping

Validation set results with adaptive clipping are shown in Figure 5 and with fixed clipping in Figure 6. These charts show that we have identified the noise regime in which performance is beginning to degrade. There is always a tension between privacy and utility: as the amount of noise increases, eventually performance will go down. For the purpose of this study we consider more than a 5% relative reduction in evaluation metric to be unacceptable. Therefore for each task, we look at the level of noise $z^*$ at which the evaluation metric on the validation set is still within 5% of the value with no noise, but adding more noise would degrade performance beyond 5%. Given $z^*$ for each task, we then choose $C^*$ to be the fixed clip value that gives best performance on the validation set. The values of $z^*$ and $C^*$ are shown in Figure 7 (left).

For our final test set evaluation, we compare adaptive clipping to the median to fixed clipping at $C^*$. The results are in Figure 7 (right). We show the average test set performance and bootstrapped 95% confidence interval over 20 runs varying the random seed used for client selection and DP noise. On three of the tasks (CIFAR-100, EMNIST-AE, SHAKESPEARE), clipping to the median actually outperforms fixed clipping to the best fixed clip chosen in hindsight, and on two more (EMNIST-CR, SO-NWP), the performance is comparable. Only on SO-LR the best fixed clip does perform somewhat better. This task seems to be unusual in that best performance comes from aggressive clipping, so a small fixed clip fares better than adaptive clipping to the median. However,

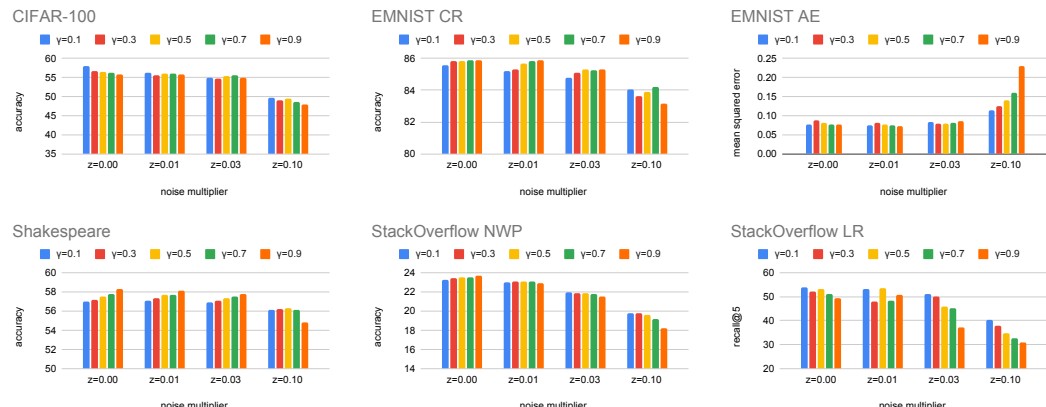

Figure 5: Evaluation metric performance of adaptive clipping with five settings of $\gamma$ for each of four effective noise multipliers $z$. Note that the $y$-axes have been compressed to show small differences, and that for EMNIST-AE lower values are better.

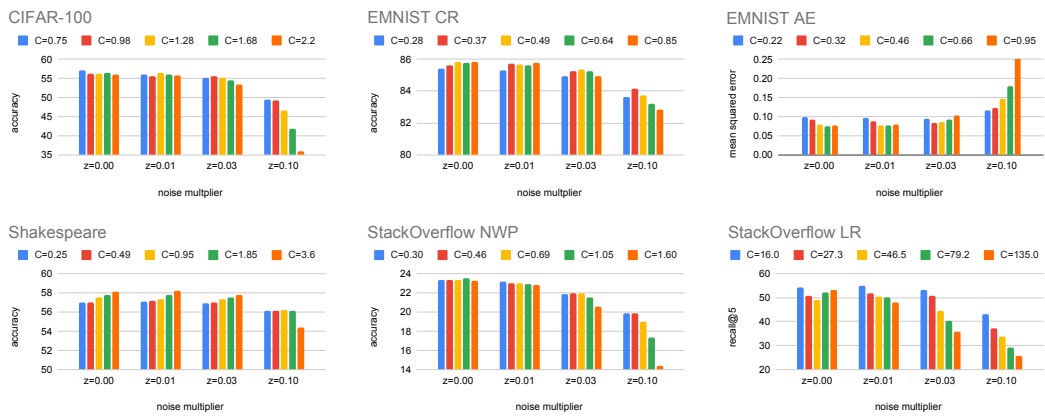

Figure 6: Evaluation metric performance of fixed clipping with five settings of $C$ for each of four noise multipliers $z$. Note that the $y$-axes have been compressed to show small differences, and that for EMNIST-AE lower values are better.

looking at Figure 6 (and noting the scale of the $y$ axis), on this task more than the others, getting the exact right fixed clip is important. The development set recall@5 value of 55.1 corresponds to the optimal fixed clip of 16.0. The next larger fixed clip of 27.3 gave a recall of only 51.8, and larger clips fared even worse. So an expensive hyperparameter search may be necessary to even get close to this high-performing fixed clip value.

## 4 Conclusions and implications for practice

In our experiments, we started with a high-performing non-private baseline with optimized client and server learning rates. We then searched over a small grid of larger server learning rates for our experiments with clipping (adaptive or fixed). This is one way to proceed in practice, if such non-private baseline results are available. More often, such baseline learning rates are not available, which will necessitate a search over client learning rates as well. In that case, it would be beneficial to enable adaptive clipping to the median *during* that hyperparameter search. The advantage of clipping relative to the unclipped baseline observed on some tasks could only increase if the other hyperparameters such as client learning rate were also chosen conditioned on the presence of clipping.

Although the experiments indicate that adaptive clipping to the median yields generally good results, on some problems (like SO-LR in our study) there may be gains from tuning the target quantile. It

| Task | $z^*$ | $C^*$ |
|------|-------|-------|
| CIFAR-100 | 0.03 | 0.98 |
| EMNIST-CR | 0.10 | 0.37 |
| EMNIST-AE | 0.03 | 0.32 |
| SHAKESPEARE | 0.10 | 0.95 |
| SO-NWP | 0.01 | 0.30 |
| SO-LR | 0.01 | 16.0 |

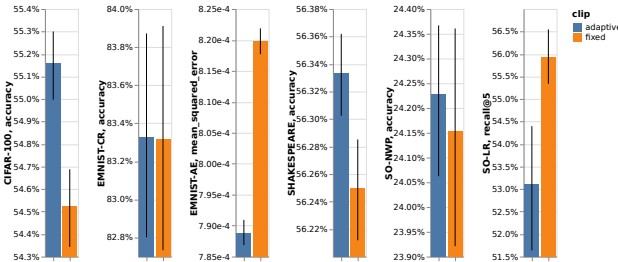

Figure 7: Left: for each task, the maximum noise possible before performance begins to significantly degrade ($z^*$), and the best fixed clip ($C^*$) chosen on the development set. Right: average test set performance and bootstrapped 95% confidence interval over 20 runs. In practice, finding the best fixed clipping norm would require substantial additional hyperparameter tuning.

would require adding another dimension to the hyperparameter grid, exponentially increasing the tuning effort, but even this would be preferable to tuning the fixed clipping norm from scratch, since the grid can be smaller: we obtained good results on all problems by exploring only five quantiles, but the update norms in the experiments range over four orders of magnitude, from 0.22 to 135.

Combining our results with the lessons taken from [18] and [24], the following strategy emerges for training a high-performing model with user-level differential privacy. We assume some non-private proxy data is available that may have comparatively few users $n'$, as well as that the true private data has enough users $n$ that the desired level of privacy is achievable.

1. With adaptive clipping to the median enabled, using a relatively small number of clients per round ($m \approx 100$), and a small amount of noise ($z = 0.01$), search over client and server learning rates on non-private proxy data.

2. Fix the client and server learning rates. Using the non-private data and low value of $m$, train several models increasing the level of noise $z$ until model performance at convergence begins to degrade.

3. To train the final model on private data, if $(m, z)$ is too small for the desired level of privacy even given $n$, set $(m, z) \leftarrow \alpha \cdot (m, z)$ for some $\alpha > 1$ such that the privacy target $(\epsilon, \delta)$ is achieved.[5] Finally, train the private model using that value of $m$ and $z$.

By eliminating the need to tune the fixed clipping norm hyperparameter which interacts significantly with the client and server learning rates, the adaptive clipping method proposed in this work exponentially reduces the work necessary to perform the expensive first step of this procedure.

---

[5]Here we employ our assumption that $n$ is sufficiently large, since $m$ cannot exceed $n$.

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
