| Task | $z^*$ | $C^*$ | adaptive | fixed |
|---|---|---|---|---|
| CIFAR-100 | 0.03 | 0.98 | 55.1 | 55.1 |
| EMNIST-CR | 0.10 | 0.37 | 85.0 | 84.8 |
| EMNIST-AE | 0.03 | 0.32 | 0.080 | 0.083 |
| SHAKESPEARE | 0.10 | 0.95 | 56.4 | 56.3 |
| SO-NWP | 0.01 | 0.30 | 24.4 | 24.4 |
| SO-LR | 0.01 | 16.0 | 51.5 | 56.1 |

Table 2: For each task, with the maximum noise possible before performance begins to significantly degrade ($z^*$), the best fixed clip ($C^*$) chosen on the development set, and the test set performance of adaptive clipping to the median compared to fixed clipping to $C^*$. In practice, finding the best fixed clipping norm would require substantial additional hyperparameter tuning.

| Task | adaptive $(\mu, \sigma)$ | | fixed $(\mu, \sigma)$ | |
|---|---|---|---|---|
| CIFAR-100 | 54.8 | 0.50 | 54.5 | 0.40 |
| EMNIST-CR | 83.3 | 1.2 | 83.3 | 1.3 |
| EMNIST-AE | 0.0804 | 0.0016 | 0.0820 | 0.00050 |
| SHAKESPEARE | 56.3 | 0.88 | 56.2 | 0.89 |
| SO-NWP | 24.2 | 0.43 | 24.2 | 0.49 |
| SO-LR | 54.5 | 2.9 | 55.9 | 1.5 |

Table 3: For each task, the mean ($\mu$) and standard deviation ($\sigma$) test set performance of adaptive clipping to the median compared to fixed clipping to the optimal $C^*$ over 20 independent runs. The randomness comes from both the Gaussian noise added for DP and the selection of clients for each round.

# A  Appendix

To be more confident that the results of Table 2 (duplicated here for easy comparison) were not due to chance, we repeated the final test experiments 20 times for each task and clipping style, varying the random seed used to select clients at each round and to generate the Gaussian noise for differential privacy. The mean and standard deviation of the evaluation metrics as computed on the test set are shown in Table 3.

We observe that adaptive clipping to the median performs comparably to fixed clipping to the optimal norm chosen in hindsight on all of the tasks. Indeed, this experiment demonstrates that some of the large performance gap observed on the SO-LR task from Table 2 is likely due to the high variance.