# OpenReview forum: "Differentially Private Learning with Adaptive Clipping"
_NeurIPS.cc/2021/Conference — NeurIPS 2021 Poster_

### Official Review · Reviewer_nrc3 · 2021-07-12

**Rating:** 6
**Confidence:** 3

**Summary:**

This works studies the problem of training neural networks with user-level differential privacy using federated learning. To bound user contribution, the authors propose to clip the gradient norms to the private estimate of the median of gradient norms, as opposed to some fixed clipping threshold. This method reduces the need to tune the clip threshold and achieves competitive performance across a range of tasks.

**Limitations And Societal Impact:**

Yes.

**Main Review:**

This work proposes to clip to the median of the gradient norms to bound user contributions in federated learning under user-level DP. The authors run extensive experiments to show that clipping to the median yields desired performance comparable to clipping at the best threshold chosen in hindsight. The experiment setup is thorough, and the results support the claims of the work.

The algorithm to estimate the median (and arbitrary quantiles) is interesting. It does not require prior knowledge of the upper bound on the gradient norm. However, it is unclear the advantage of this method over existing private quantile estimators, for example,

Dwork and Lei, Differential Privacy and Robust Statistics

Tzamos et.al., Optimal Private Median Estimation under Minimal Distributional Assumptions.

The authors may want briefly discuss why the current algorithm is preferred over existing algorithms for quantile estimation.

Another issue is the privacy guarantees. It seems that in the experiments the authors choose $\sigma_b, \sigma_\Delta$ that guarantees $(5, n^{-1.1})$-DP, which falls into the low privacy regime. The authors provided a procedure to tune the parameters to satisfy arbitrary privacy guarantees, but it is unclear to me when we should stop increasing the noise levels. The authors may wish to provide a more formal privacy guarantee in terms of the parameters (e.g., clip threshold, noise magnitudes).

Overall the paper is well-organized.

**Time Spent Reviewing:**

4

---

> ### Author Response · Authors · 2021-08-10
> **Response to review.**
>
> Thank you for your review.
> * The main advantage of our approach relative to the other quantile estimation algorithms you mentioned is that it lets us track the moving quantile very efficiently (with respect to privacy budget), and transmitting the bare minimum of information from client to server. The methods you mentioned would estimate the quantile afresh at each round, which would entail using much more privacy budget. We can get by with an almost negligible impact on privacy essentially by making use of information across multiple rounds.
> * Regarding when to stop increasing noise levels. Our stated goal was to stop increasing noise levels when the accuracy falls below 5% of the unnoised level. This lets us train the final model to meet the desired level of privacy with predefined accuracy impact, minimizing computational resources (clients per round). If tighter privacy guarantees were desired, it would be possible to train with more noise and proportionally more clients per round, which would increase privacy at the expense of more computation (while accuracy should be essentially unchanged). This is similar to the approach advocated by [[McMahan et al., 2018]](https://arxiv.org/abs/1710.06963), except that we allow for a predefined potentially non-zero accuracy impact.

---

### Official Review · Reviewer_Z3R2 · 2021-07-15

**Rating:** 6
**Confidence:** 4

**Summary:**

This paper tackles the problem of training accurate neural networks with user-level privacy in the federated learning settings. Classical implementations usually ensure user-level differential privacy of the learning scheme by applying the following procedure:

1) before sending their gradient update to the server, every user clip (or rescale) its gradient update so that its norm is smaller or equal to a constant clipping parameter $C$
2) after aggregating the clipped gradients, the server adds Gaussian noise proportional to $C$ before applying a gradient update rule.

In this paper, instead of applying step 1) with a fixed constant $C$, the authors present a method to adaptively select the clipping constant during the training. This method is based on a quantile regression problem under differential privacy constraints. The authors demonstrate through numerical experiments that this new adaptive clipping method obtain similar result as classical implementations while eliminating the need for hyper parameter search on $C$.

**Limitations And Societal Impact:**

I don't see any negative social impact rising from this paper.

**Main Review:**

Relevance: Federated Learning is currently a hot topic within the ML community. Furthermore, improving the convergence/accuracy of privacy preserving techniques is also very important and relevant to the NeurIPS community.

Clarity: The paper is overall well written and statements are clearly presented.

Quality: Both the technical and experimental part of the paper look sound to me. I just have some minor comments/questions.
- In the proof of Theorem 1, why are you building a query that sends $(\frac{\Delta_i^t}{\sigma_\Delta},\frac{(b_i^t -1/2)}{\sigma_b})$ instead of $(\frac{\Delta_i^t}{\sigma_\Delta},\frac{b_i^t }{\sigma_b})$ ? Also why is $\frac{1}{2\sigma_b}$ the sensitivity of $(b_i^t-1/2)/\sigma_b$ and not $1/\sigma_b$?
- Figure 2 is interesting as it presents a sanity check of the quantile regression problem for a fixed gradient distribution. However it is unclear to me how informative it is in the context of the gradient descent, as the gradient distribution will be changing at every step.
- I am not aware of the classical privacy level the literature on Private FL considers as acceptable, but, from my point of view, $\epsilon=5$ is not a very meaningful privacy budget. Maybe it would have been interesting to compare the privacy/accuracy tradeoff for several privacy budgets.

Originality and Significance: I think originality is not the main strength of this paper. The idea of using the median to adapt the gradient clipping during the learning was first presented in [1], even though it was not properly studied. Similarly, the problem of quantile regression is a plain old machine learning problem and adapting it with differential privacy only asked for Gaussian noise injection during SGD. Nevertheless, I think the idea of combining these techniques in a sound way to build efficient adaptive clipping is new. Overall, while the idea is not very surprising, I think this is an interesting paper that could benefit to the community by helping automatize the study of differentially private implementations of SGD.


**Time Spent Reviewing:**

6-7 hours

---

> ### Author Response · Authors · 2021-08-10
> **Response to review.**
>
> Thank you for your review.
> * The transformation described in the proof of Theorem 1 has two purposes. First, it makes the noise added to the combined update-and-clipped-indicator spherical so we can analyze it as a single application of the Gaussian mechanism. Second, translating the clipped count indicator by -½ reduces its sensitivity from 1 to ½. For the sensitivity analysis, the key fact is that the norm of the concatenated vector is bounded by S. In particular, since $b_i^t \in [0, 1]$, we have $|b_i^t -½| \in [-½, ½]$, so $(b_i^t - ½)/\sigma_b \in [-1/(2\sigma_b), 1/(2\sigma_b)]$.
> * We include Figure 2 specifically so that we can visualize how well the algorithm tracks a known quantile. Fig 4. conflates the error of the quantile matching algorithm and the changing, unknown, true quantile value.
> * Regarding the choice of $\epsilon=5$ in table 1. There have been prior works in the literature that demonstrate using DP-FedAvg with moderate values of epsilon significantly reduces unintended memorization in trained models. For instance, [[Thakkar et al. 2021]](https://arxiv.org/abs/2006.07490) shows this in Language Models with DP-FedAvg having eps=18.8, and [[McMahan et al., 2018](https://arxiv.org/abs/1710.06963),[Ramaswamy et al. 2020](https://arxiv.org/abs/2009.10031)] show training Language Models with DP-FedAvg that have eps={4.63, 5.36} in simulated settings. However, it is important to note that existing techniques are able to achieve competitive utility with the stated privacy levels only at significantly increased computation costs (~1.5-2 orders of magnitude higher in [[McMahan et al., 2018](https://arxiv.org/abs/1710.06963),[Ramaswamy et al. 2020](https://arxiv.org/abs/2009.10031)]). Our value of $\epsilon = 5$ could also be made smaller at the expense of more computation (clients per round).

---

### Official Review · Reviewer_BgsA · 2021-07-16

**Rating:** 4
**Confidence:** 4

**Summary:**

This paper studies the problem of ERM under user-level differential privacy. They propose a method based on quantile clipping to bound the contribution of each user (L_2 sensitivity of the gradient estimation), eliminating the need to specify a clipping threshold in advance.

**Limitations And Societal Impact:**

See above.

**Main Review:**

Strengths:
- The problem of ERM under user-level differential privacy is important and has real-world applications.
- To bound the contribution of each user, this paper chooses the clipping threshold by an adaptive method based on quantile, instead of requiring a pre-specified clipping threshold.

Weaknesses:
- The novelty and technical depth is low: the quantile computation method directly follows from a standard method based on gradient descent; there is no theoretical analysis for the accuracy guarantee (e.g. for the quantile computation), and the heuristic of clipping to median is proposed in DP-SGD.
- For quantile computation under differential privacy, the comparison with smooth sensitivity is not accurate and complete. At lines 72-73, the authors say that smooth sensitivity requires each client send unclipped updates to the server, however, only their norms are needed to estimate the median. In fact, since the privacy guarantee is in the centralized model, this paper also requires the server knowns the exact answers (i.e., the information of unclipped updates to estimate the median, and the aggregation of all client updates) to be perturbed by Gaussian noise.
- Lack of clear statements (e.g., theorems and proofs) for the privacy and accuracy guarantee. It is not easy to follow and understand the details of the proposed quantile computation method.

[1] Answering Range Queries Under Local Differential Privacy


**Time Spent Reviewing:**

1

---

> ### Author Response · Authors · 2021-08-10
> **Response to review.**
>
> Thank you for your review.
> * *Communication requirements of estimating the median with smooth sensitivity*. Our method transmits significantly less information than the full update, because the update is clipped before transmission. If we used smooth sensitivity we’d have to *either* send the unclipped updates *or* send the true update norms in one round of communication and then the clipped updates in a second round. The server therefore receives strictly less information with our approach. Indeed, unlike smooth sensitivity, our approach is compatible with distributed differential privacy in which the server is essentially oblivious [[Kairouz et al., 2021]](https://arxiv.org/abs/2102.06387).
> * *Lack of clear statements and difficulty of understanding quantile estimation method*. We can make the statement of the theorem more precise try to restructure the exposition so a verbal description precedes the mathematical description in section 2. If anything specific is unclear, please let us know.

---

### Official Review · Reviewer_qHtK · 2021-07-21

**Rating:** 5
**Confidence:** 3

**Summary:**

This paper studies the problem of how to choose a good clipping norm in differentially private federated learning. The authors propose a new method to select the clipping threshold by privately approximate the quantile of the update norm distribution. Experiments validate the effectiveness of the proposed method.

**Limitations And Societal Impact:**

Yes

**Main Review:**

The idea of the proposed method seems to be interesting, but the analysis of the current method needs to be further justified. Here are my main concerns of the current paper:
1. The motivation of using the quantile of the update distribution as the clipping threshold is not well justified.
2. The privacy guarantee of the current method is not very clear. For example, how the number of sampled clients will affect the privacy guarantee is not clear. In addition, it is not clear how the proposed method tracks the privacy loss in the experiments.
3. It is unclear how the proposed method will affect the utility guarantee of the algorithm. The contribution of the current paper will be stronger if the authors could add more analysis about the utility using the proposed clipping method.
4. One important metric to evaluate the performance of federated learning is the communication round. How the proposed method will affect the communication round is also unclear.
5. Experimental results show that the improvements of the proposed methods are not very significant compared with the fixed clipping method.

**Time Spent Reviewing:**

8

---

> ### Author Response · Authors · 2021-08-10
> **Response to review**
>
> Thank you for your review.
> * *Privacy guarantee*. Our Theorem 1 shows an iteration to be equivalent to a single application of the Gaussian mechanism. Therefore any of the many methods to estimate privacy of repeated compositions of the subsampled Gaussian mechanism apply to our method. In particular, we use the method described in the paper “Subsampled Rényi differential privacy and analytical moments accountant” by Wang, et al. (cited in multiple places in the paper) which is implemented [here]( https://github.com/tensorflow/privacy/blob/master/tensorflow_privacy/privacy/analysis/rdp_accountant.py#L397).
> * *The effect of our approach on utility*. Our benchmark tasks are (other than SO-LR) non-convex optimization problems for which utility guarantees are not known. However our experiments (e.g., Fig. 3) demonstrate that clipping the updates to the estimated median norm does not significantly affect utility. Indeed on three of our six test problems, it helps, perhaps by reducing the impact of outlier clients.
> * *Communication round cost*. As shown in the function “FedAvg” in Algorithm 1, our method requires a single extra bit to be communicated by each client, in addition to their clipped update. As practical models usually have millions of parameters, that extra communication cost can be considered to be negligible.
> * *Improvements relative to fixed clipping*. It is critical to note that the fixed clipping baseline was given the benefit of choosing the best fixed clipping norm in hindsight, making it artificially strong. In order to find the best clipping norm for all of the studied problems it would be necessary to search over a range of clip values that spans several orders of magnitude. We do not claim our adaptive clipping method performs *better* than fixed clipping, only that it performs comparably without having to fit any clipping norm hyperparameter: we clip to the estimated median on all problems, starting from the same initial estimate.

---

### Decision · Program_Chairs · 2021-09-27

**Decision:**

Accept (Poster)

**Comment:**

The paper presents a method for adaptively setting the gradient clipping threshold for DP federated learning. The proposed method is mostly a combination of known techniques, but it provides a neat and well-presented solution to an important practical problem.

The reviews are mixed: some reviewers are more positive because of more emphasis on the practical relevance while others are more negative because of emphasis on limited technical novelty.

The AC is in favour of accepting the paper because of its practical usefulness and potential impact. In my opinion, the authors have been able to answer the criticisms of the negative reviews in their rebuttal. Unfortunately these reviewers have been unresponsive in the discussion and thus the reviews have not been updated in light of the author responses, potentially leaving the reviews unnecessarily negative.

While not reflected in the reviews, this paper was discussed extensively by the AC (who read it in full and championed the work) and the SAC, leading to the decision to accept.